# Epigenetic Associations between lncRNA/circRNA and miRNA in Hepatocellular Carcinoma

**DOI:** 10.3390/cancers12092622

**Published:** 2020-09-14

**Authors:** Tae-Su Han, Keun Hur, Hyun-Soo Cho, Hyun Seung Ban

**Affiliations:** 1Korea Research Institute of Bioscience and Biotechnology (KRIBB), Daejeon 34141, Korea; tshan@kribb.re.kr; 2Department of Biochemistry and Cell Biology, School of Medicine, Kyungpook National University, Daegu 41944, Korea; KeunHur@knu.ac.kr

**Keywords:** microRNA, long non-coding RNA, circular RNA, competing endogenous RNA, hepatocellular carcinoma

## Abstract

**Simple Summary:**

Non-coding RNAs such as microRNAs, long non-coding RNAs, and circular RNAs contribute to the development and progression of hepatocellular carcinoma through epigenetic association. Long non-coding RNAs and circular RNAs act as competing endogenous RNAs that contain binding sites for miRNAs and thus compete with the miRNAs, which results in promotion of miRNA target gene expression, thereby leading to proliferation and metastasis of hepatocellular carcinoma. Competing endogenous RNAs have the potential to become diagnostic biomarkers and therapeutic targets for treatment of hepatocellular carcinoma.

**Abstract:**

The three major members of non-coding RNAs (ncRNAs), named microRNAs (miRNAs), long non-coding RNAs (lncRNAs), and circular RNAs (circRNAs), play an important role in hepatocellular carcinoma (HCC) development. Recently, the competing endogenous RNA (ceRNA) regulation model described lncRNA/circRNA as a sponge for miRNAs to indirectly regulate miRNA downstream target genes. Accumulating evidence has indicated that ceRNA regulatory networks are associated with biological processes in HCC, including cancer cell growth, epithelial to mesenchymal transition (EMT), metastasis, and chemoresistance. In this review, we summarize recent discoveries, which are specific ceRNA regulatory networks (lncRNA/circRNA-miRNA-mRNA) in HCC and discuss their clinical significance.

## 1. Introduction

Hepatocellular carcinoma (HCC) is one of the most commonly diagnosed cancers and highly lethal malignancies. There have been 42,810 new cases and 30,160 deaths in the United States in 2020 [1]. Risk factors for HCC are hepatitis B virus (HBV) or hepatitis C virus (HCV) infection, alcohol consumption, nonalcoholic fatty liver disease (NAFLD), non-alcoholic steatohepatitis (NASH), and hereditary diseases [2,3,4,5,6,7]. HCC patients are treated with potentially curative resection in approximately 30–40% of cases; however, these patients still have a poor prognosis because of the high frequency of metastasis and recurrence [8,9,10]. During HCC progression, cellular changes, including inflammation, hypoxia, and the tumor microenvironment are important as various molecular events can occur. There are major signal transduction pathways that promote HCC, including Myc, MAPK, PI3K, WNT, and JAK [11,12,13,14]. Despite these well-known signaling pathways, there is still inadequate information to understand HCC progression. Thus, novel target molecules are urgently required for the application of diagnostic biomarkers and therapeutic agents.

Non-coding RNAs (ncRNAs) were traditionally considered “junk genes.” Recently, various ncRNAs have been identified by developing next-generation sequencing techniques and found to play a critical role in the regulation of gene expression by binding to promoters or directly interacting with proteins [15,16,17]. Accumulating evidence has shown that ncRNAs are involved in normal cellular processes, but their dysregulation is associated with disease progression, including cancer [18,19]. Furthermore, dysregulated non-coding RNAs are associated with HCC initiation, progression, and metastasis [20].

Long ncRNAs are longer than 200 nucleotides, generally do not code for proteins, and function as master regulators. Numerous studies have revealed the biological contributions of lncRNAs as regulators of transcription, modulators of mRNA processing, and organization of nuclear domains [21,22]. However, dysregulated lncRNAs are involved in the pathological processes of cancers, including cell growth, survival, and differentiation by functioning as oncogenes or tumor suppressors [23,24].

Circular RNAs are endogenous ncRNAs that lack 5′ and 3′ ends and are products of backsplicing on precursor mRNAs [25]. CircRNAs are evolutionarily conserved and have high stability because of their circular structure; thus, they are inherently resistant to RNase activity. In addition, several studies have shown that there are specific miRNA binding sites in circRNA sequences. Therefore, recent studies have focused on the ability of miRNA sponges to regulate gene expression. Furthermore, increasing evidence shows that aberrant expression of circRNAs can be mediated in cancer progression due to its important biological function, as a miRNA sponge [26,27].

Short ncRNAs include microRNA (miRNA), small interfering RNA (siRNA), snoRNA, rRNA, tRNA, and Piwi-interacting RNA (piRNA) [28,29]. Among them, most studies have focused on miRNAs. Most miRNAs are transcribed by RNA polymerase II; however, small groups of miRNAs are transcribed by polymerase III. Micro RNAs are single-stranded RNAs and play a role as negative gene regulators by base pairing to partially complementary sites on the target mRNA 3′-untranslated region (UTR) [30,31]. When miRNAs bind to target mRNAs, target genes undergo translation repression or decay (Figure 1) [32,33]. A single miRNA can regulate multiple targets containing specific miRNA response elements (MREs). In addition, a single RNA contains multiple MREs; therefore, multiple miRNAs can regulate a single RNA [34]. Abnormal expression of miRNAs can affect cancer development, including cell proliferation, angiogenesis, apoptosis, and cell motility [35,36,37,38,39,40]. Several researchers have found that molecular mechanisms influence carcinogenesis [41,42]. There are two types of miRNAs that play roles in cancer (including HCC), tumor suppressor miRNAs and onco-miRs [43,44,45].

In the past few years, most miRNA studies have focused on the unidirectional regulation of target transcripts; however, competitive miRNA binding has been observed using artificial miRNA sponges, which act as inhibitors of multiple miRNAs [46,47,48]. The first natural miRNA sponge was identified in *Arabidopsis thaliana*, in which it sequestered miR-399 and inhibited its activity by “target mimicry.” Most miRNA targets are cleaved by miRNAs in plants owing to their almost perfect miRNA match. However, the miR-399 motif on IPS1 contains a mismatched loop at the miRNA binding site that eliminates cleavage. Therefore, IPS1 can act as a miR-399 sponge and change the stability of its target, PHO2 mRNA [49,50]. In animal cells, Ebert et al. observed similar phenomena: artificial overexpression of miRNA binding sites leads to upregulation of miRNA targets acting as RNA sponges [51]. Since 2011, this kind of post-transcriptional regulation has been described by the “competing endogenous RNA (ceRNA)” model, which describes competitive binding between sponge RNA and miRNA target genes and regulation of miRNA target gene activity (Figure 2) [52]. Importantly, numerous studies have shown that lncRNAs or circRNAs can act as ceRNAs containing miRNA-binding sites. Therefore, miRNAs can be suppressed by increasing the stability of miRNA target mRNAs. Notably, this ceRNA mechanism has been discovered in diverse diseases, including multiple cancers [53,54].

Understanding the role of ncRNAs in tumorigenesis is a major challenge in recent molecular oncology. Therefore, in this review, we introduce the mechanism of the ceRNA network between lncRNA/circRNAs and miRNAs and discuss its possible role in HCC progression.

## 2. Long Non-Coding RNA and microRNA Networks in HCC

Although most lncRNA functions are unknown, several functional studies have shown that they are closely associated with cancer progression. Growing evidence indicates that their tumorigenicity may be mediated by ceRNA regulatory mechanisms. In this section, we present the details of the regulatory network between miRNAs and lncRNAs in HCC from recent and prominent studies (Table 1).

### 2.1. Overexpression of lncRNAs Promotes HCC Proliferation and Metastasis via Sponging of Tumor Suppressing miRNAs

In a lncRNA microarray assay, SNHG11 was overexpressed and associated with poor prognosis in HCC. To regulate HCC proliferation, SNHG11 was negatively regulated by miR-184, which directly targets *AGO2*. In HCC tissues, SNHG11 was negatively correlated with miR-184 and positively correlated with *AGO2* expression [55]. The lncRNA CCAT1 was overexpressed in HCC tissues and sponged miRNA let-7 leading to upregulation of *HMGA2* and *c-MYC* expression [56]. To regulate the cell cycle, lncRNA SNHG16 absorbed miRNA let-7b-5p, and SNHG16 promoted the G2/M transition via regulation of the let-7b-5p/*CDC25B* axis [57]. In addition, knockdown of lncRNAs HOXA-AS2 and CDKN2B-AS1 induced cell apoptosis via G1 arrest. HOXA-AS2 sponged miR-520c-3p and let-7c-5p, and upregulated *GPC3* and *NAP1L1* expression by downregulating miR-520c-3p and let-7c-5p [26,58]. In addition, overexpression of FEZF1-AS1 and H19 in HCC sponged miR-4443 and miR-326 leading to HCC growth and metastasis, respectively. MiR-326 directly targeted the transcription factor TWIST. Subsequently, downregulation of miR-326 by H19 induced *TWIST* expression leading to HCC development and metastasis [59,60].

In HCC metastasis, high levels of MALAT1 and FOXD2-AS1 increase *vimentin* and *ANXA2* expression by sponging miR-30a-5p and miR-206, respectively. In a migration/wound healing assay, siMALAT1 and siFOXD2-AS1 treatment reduced the migration and wound healing rate compared to the siNegative control by reduction of *TWIST1*/*ANXA2* expression and upregulation of miR-30a-5p and miR-206 [61,63]. In addition, lncRNA MALAT1 sponged miR-146b-5p to induce *TRAF6* expression leading to HCC metastasis [62]. Moreover, lncRNA TINCR is a sponge for miR-214-5p. TINCR overexpression sponged miR-214-5p to upregulate *ROCK1* in HCC metastasis [64]. Knockdown of lncRNA SNHG15 and SNHG8 suppressed HCC metastasis and proliferation via regulation of the miR-490-3P/ *HDAC2* axis and miR-149-5p/ *PPM1F* axis, respectively. *HDAC2* and *PPM1F* were direct targets of miR-490-3P and miR-149-5p, and overexpression of SNHG15 and SNHG8 in HCC showed a correlation between *HDAC2* and *PPM1F* expression via the absorption of miR-490-3P and miR-149-5p, respectively [65,66]. The lncRNA FLVCR1-AS1 sponged miR-513c to modulate HCC metastasis and proliferation via up-regulation of *MET* expression [67]. In addition, overexpression of lncRNA *ROR* induced *ZEB2* expression by sponging miR-145, and increased EMT and HCC metastasis [68]. The lncRNA MIAT sponged miR-520d-3p, upregulating *EPHA2* expression leading to HCC proliferation and metastasis [81]. In addition, knockdown of LINC00460 and LINC00488 induced cell apoptosis and reduced angiogenesis via downregulation of *PAK1* and *TLN1*, respectively, which are direct targets of miR-485-5p and miR-330-5p. LINC00460 and LINC00488 sponged miR-485-5p and miR-330-5p leading to HCC tumorigenesis and angiogenesis, respectively [69,70].

To regulate the HCC signaling pathway, lncRNA DSCR8 acts as a sponge for miR-485-5p and regulates the Wnt/β-catenin signaling pathway resulting in upregulation of *FZD7*. Statistical analysis of DSCR8 and miR-485-5p showed a close relationship between malignant clinicopathological features and survival rate [71]. In addition, lncRNAs DBH-AS1 and TUG1 up-regulated the FAK/Src/ERK and JAK2/STAT3 pathways by sponging miR-138 and miR-144 resulting in HCC tumorigenesis [72,73]. The lncRNAs SNHG12 and SNHG6-003 upregulated the NF-κB and p38 pathways via induction of *MLK3* and *TAK1* expression by sponging miR-199a/b-5p and miR-26a/b, respectively [74,75]. Moreover, lncRNA UCA1 activated the FAFR1/ERK signaling pathway by regulating *FGFR1* expression by sponging miR-216b [76].

During autophagy, upregulation of lncRNAs NEAT1 and CCAT absorbed miR-204 and miR-181a-5p to induce HCC autophagy via upregulation of *ATG3* and *ATG7*, respectively [77,78]. In addition, lncRNA PVT1 induced HCC autophagy via regulation of the miR-365/*ATG3* axis. Overexpression of PVT1 sponged miR-365 in HCC; consequently, *ATG3* expression was increased by HCC autophagy induction [79].

Moreover, lncRNA MIAT is associated with senescence in HCC. Knockdown of MIAT induced cellular senescence and HCC growth. The target of miR-22-3P is Sirtuin 1 (*SIRT1*), and overexpression of MIAT downregulated miR-22-3P via the sponge effect, and *SIRT1* expression increased. Downregulation of MIAT resulted in senescence-associated secretory phenotype and suppressed HCC tumorigenesis [80]. Overall, overexpressed lncRNAs are critically involved in HCC proliferation and metastasis via regulation of the cell cycle, autophagy, apoptosis, and several signaling pathways. Thus, overexpressed lncRNAs are recognized as HCC biomarkers and therapeutic targets.

### 2.2. Tumor Suppressor lncRNAs Inhibit HCC Tumorigenesis by Sponging Onco-miRNAs

In HCC cell lines and tissues, lncRNA GAS5 expression decreased. Knockdown of GAS5 induced doxorubicin resistance and promoted cell proliferation via upregulation of *PTEN*. Although miR-21 directly downregulated *PTEN* expression in HCC, overexpression of GAS5 was sponged by miR-21. Consequently, *PTEN* expression increased and inhibited HCC [82]. In addition, lncRNA SNHG16 overexpression inhibited HCC proliferation and 5-fluorouracil (5-FU) chemoresistance in in vivo/in vitro assays via absorption of has-miR-93 in Hep3B and Huh7 cell lines [83]. The lncRNA XIST was downregulated in HCC. Upregulation of miR-497-5p inhibits the expression of *PDCD4* (programmed cell death 4). In a cell growth assay, overexpression of XIST inhibited the growth of HepG2 cell lines via upregulation of *PDCD4* and absorption of miR-497-5p [84]. Downregulation of lncRNA EPB41L4A-AS2 in HCC is clearly associated with negative regulation of HCC proliferation and metastasis. Overexpression of EPB41L4A-AS2 sponged miR-301a-5p and inhibited cell growth and migration/invasion via upregulation of *FOXL1* by miR-301a-5p downregulation. In an in vivo study, EPB41L4A-AS2 suppressed lung metastasis via regulation of the miR-301a-5p/*FOXL1* axis [85]. The expression of lncRNAs DGCR5 and MIR31HG was negatively correlated with HCC proliferation and metastasis. Overexpression of DGCR5 and MIR31HG sponged miR-346 and miR-575, and suppressed HCC cell growth and migration/invasion via upregulation of *KLF14* and *ST7L* expression, respectively [86,87]. In addition, LINC00657 and TUSC7 were positively correlated with *PTEN* and *EPHA4* expression in HCC. Overexpression of LINC00657 and TUSC7 suppressed HCC proliferation, migration, and invasion by sponging of miR-106a-5p and miR-10a, and upregulation of *PTEN* and *EPHA4* expression [88,89]. Collectively, the study of tumor suppressor lncRNAs in HCC helps to provide an understanding of HCC proliferation and metastasis, and lncRNAs can be used as potential biomarkers for HCC diagnosis.

## 3. Circular RNA and microRNA Networks in HCC

An increasing number of studies have revealed that circRNAs play important roles in cancer development, including HCC. CircRNA can be used as a ceRNA to decrease cytoplasmic levels of target miRNAs by abolishing miRNAs. Thus, gene expression levels of target mRNAs can be maintained. In this section, we describe how individual circRNAs may participate as ceRNAs in the regulatory network of HCC.

### 3.1. Overexpression of Circrnas Induces HCC Progression by Sponging Tumor-Suppressive miRNAs

To date, several studies have found that circRNAs exert oncogenic effects by sponging miRNAs in HCC progression. In HCC tissues and cell lines, circ-PVT1 derived from the *PVT1* gene locus is markedly elevated and acts as a sponge of miR-3666 [90]. Knockdown of circ-PVT1 resulted in the reduction of proliferation and induction of apoptosis by upregulating miR-3666 in HCC cells. A molecular mechanism has been identified in which circ-PVT1 induces HCC proliferation by enhancing *SIRT7*, a target gene of miR-3666. In addition, various miRNAs including miR-125, miR-145, and miR-497 have been reported as sponge targets of circ-PVT1 in gastric cancer [91], colorectal cancer [92], and non-small cell lung cancer [93], respectively.

A study by Xiao et al. demonstrated that circRNA plays a role in the estrogen receptor (ER) α-mediated decrease of HCC cell invasion [94]. ERα inhibited circ-SMG.172 expression by binding to the 5′ promoter region, and expression of the tumor suppressor miR-141-3p increased by disruption of the sponging function of circ-SMG.172. miR-141-3p subsequently suppressed the expression of *Gelsolin* by binding to the mRNA 3′UTR. In HBV-related HCC, circ-100338 also acts as a sponge for miR-141-3p [95]. In silico analysis suggests that *MTSS1* is a potential target of miR-141-3p that regulates metastasis of HCC.

A bioinformatics analysis study found that upregulated circRNA and downregulated miRNA in HCC provided information about miRNA sponging circRNA [96]. From the qRT-PCR validation, has-circ-0009910 was found to be a sponge of miR-1261, which resulted in the enhancement of *UBE2L3* expression and HCC progression.

In the early stages of HCC, various circRNAs have been identified as important regulators involved in HCC progression. Circ-CDYL is upregulated in early stage HCC and induces expression of *HDGF* and *HIF1AN* by sponging miR-892a and miR-328-3p, respectively [97]. Circ-CDYL-induced *HDGF* activates PI3K-AKT signaling by binding to its receptor NCL, which results in enhanced expression of c-MYC and survivin. Circ-CDYL-mediated *HIF1AN* also upregulates survivin expression via inhibition of Notch2 signaling.

In HCC cells, circ-PRKCI functions as a miRNA-545 sponge and disrupts its inhibitory activity against the E2F transcription factor 7 (E2F7) [98]. Higher expression of *E2F7* is observed in HCC and correlated with a lower survival rate. In HCC, enhanced expression of has-circ-0078710 is correlated with the promotion of cell proliferation, migration, and invasion. It was found that has-circ-0078710 functions as a sponge of miR-31, which results in the induction of *HDAC* and *CDK2* target genes [99]. In HCC tissues, the upregulated expression of circ-001569 and circ-0005075 is correlated with increased HCC proliferation and metastasis. It was hsown that circ-001569 and circ-0005075 act as sponges of miR-411-5p and miR-432-5p [100], and miR-431 [101], respectively. However, their target genes in HCC have not been mentioned.

A study by Bai et al. reported that circ-FBLIM1 functions as a competing endogenous RNA (ceRNA) to induce HCC progression [102]. The molecular mechanism is such that circ-FBLIM1 is the sponge of miR-346, and *FBLIM1* is a direct target of miR-346. In HCC, overexpression of aquaporin 3 (*AQP3*) promotes cell proliferation and migration, and circ-HIPK3 expression is positively correlated with *AQP3* expression by sponging miR-124 [103]. Knockdown of circ-HIPK3 reduced tumor growth via the miR-124-AQP3 axis in the Huh7 xenograft model. Therefore, these circRNAs functioning as sponges for mi-RNAs could be used as a biomarker for diagnosis and as targets for HCC therapy. The circRNAs, miRNAs, and their target genes are summarized in Table 2.

### 3.2. Tumor Suppressor Circrnas Inhibit HCC Tumorigenesis by Sponging Onco-miRNAs

Several circRNAs that function as tumor suppressors in HCC have been reported. A study on HCC tumorigenesis found that circRNA is involved in the regulatory mechanism of oncogenic miR-191 [104]. In HCC Hep3B and HepG2 cells, elevated miR-191 was sponged by has-circ-0000204, and the expression of tumor suppressor *KLF6* increased via binding reduction of miR-191 to the 3′UTR region of *KLF6* mRNA.

A study by Wang et al. reported the roles of circRNA hippocampus abundant transcript 1 (circ-HIAT1) in HCC and its tumor suppressive mechanism [105]. In vitro and in vivo experiments demonstrated that circ-HIAT1-mediated upregulation of PTEN expression via miR-3171 sponging resulted in HCC cell proliferation.

In HCC tissues, circ-SETD3 is another tumor suppressive circRNA that acts as a sponge of miRNA. Circ-SETD3 reduces the proliferation of Huh7 HCC cells by sponging miR-421 and enhancing expression of its target gene, *MAPK14* [106]. In addition, overexpression of circ-SETD3 reduced tumor growth in a Huh7 xenograft mouse model.

circ-ADAMTS13 has also been identified as a tumor suppressor circRNA, and acts as a sponge of miR-484; however, its target gene is unknown [107]. Circ-MTO1 is another tumor suppressive circRNA that acts as a sponge of miRNA in HCC. Circ-MTO1 inhibits HCC progression by sponging oncogenic miR-9 to promote target gene *p21* expression [108]. Has-circ-0005986 exerts tumor suppressive effects by sponging miR-129-5p in HCC cell lines. Gene ontology analysis demonstrated that *Notch1* is the direct target gene of miR-129-5p [109]. These studies suggest that circRNAs function as tumor suppressors and by sponging oncogenic miRNA, might be good HCC biomarkers. Tumor suppressive circRNAs and target miRNAs are listed in Table 2.

## 4. Clinical Application of lncRNAs and CircRNAs as Novel Biomarkers in HCC

As described above, lncRNAs/circRNAs, as ceRNAs, are heavily involved in HCC development via diverse regulation of onco-miRNAs and/or tumor suppressive-miRNAs (Figure 3). Increasing evidence indicates that ceRNAs not only serve as biomarkers for the diagnosis of various cancers, including HCC, but are also involved in chemotherapy resistance. Through meta-analysis, several studies have demonstrated the diagnostic value of lncRNAs as biomarkers in HCC [110]. Another group has analyzed lncRNA and mRNA expression profiles obtained from The Cancer Genome Atlas and identified four lncRNAs (RP11-486O12.2, RP11-863K10.7, LINC01093, and RP11-273G15.2) that have a diagnostic potential for HCC [111]. GAS5 is a novel non-Ig partner of BCL6. Suppression of GAS5 is associated with clinicopathological characteristics in several liver dysfunctions [112,113,114]. Tumor node stage (TNM), overall survival (OS), disease-free survival (DFS), and metastasis were correlated with GAS5 expression, indicating that GAS5 can be a potential diagnostic and prognostic biomarker in HCC [115]. HULC (highly upregulated in liver cancer) is known to reduce the expression of protein kinase cAMP-activated catalytic subunit beta (PRKACB) and to increase the *HMGA2* oncogene via interaction with miR-372 and miR-186 in HCC [116,117]. In clinical analysis, HULC was detected more frequently in HCC patients with HBV, which was correlated with tumor size and tumor capsular invasion [118]. The plasma expression level of MALAT1 in HCC patients was associated with liver damage and showed clinical potential for predicting HCC development [119]. In addition, the serum level of UCA1 was higher in HCC patients [120]. Receiver operating characteristic (ROC) curve analysis revealed that serum UCA1 levels could distinguish HCC patients from healthy controls (AUC = 0.902) with high sensitivity and specificity. However, in order to apply this criterion clinically, additional evaluations are required. GALAD (which includes gender, age, AFP-L3, alpha-fetoprotein, and des-carboxy-prothrombin) and BALAD (which includes bilirubin, albumin, AFP-L3, alpha-fetoprotein, and des-carboxy-prothrombin) score calculations may be particularly useful [121,122].

As circRNAs are abundant, stable, and highly conserved, they have great potential as cancer diagnostic biomarkers in HCC. Previously, Shang et al. discovered 26 upregulated and 35 downregulated circRNAs in HCC tissues compared to adjacent non-tumorous tissues [123]. Among these 61 differentially expressed circRNAs, only hsa_circ_0005075 displayed differences in HCC, which correlated with tumor size and poor prognosis and exhibited good diagnostic potential (AUROC = 0.94) [123,124]. In addition, differentially expressed hsa_circ_0004018 and hsa_circ_0128298 were identified in HCC by circRNA microarray [125,126]. After further validation via qRT-PCR, these two circRNAs showed strong potential as novel diagnostic and prognostic biomarkers in HCC patients. More recently, 13,124 circRNAs were identified in HBV-associated liver cancer patients using bioinformatics tools after high-throughput RNA sequencing [127]. Prediction analysis for circRNA-miRNA interactions revealed that 6020 circRNAs had putative binding sites for 1654 miRNAs. One of the identified circRNAs was circRNA_10156, which is up-regulated in liver cancer and leads to enhanced Akt1 expression via miR-149-3p sponging. Therefore, circRNA_10156 may be a useful biomarker for HBV-related liver cancer diagnosis.

Owing to their high stability and abundance, circRNAs in several body fluids such as blood, urine, and saliva, are considered suitable biomarkers for liquid biopsies [128]. Although alpha-fetoprotein (AFP) is the most widely used serum/plasma biomarker for HCC diagnosis, it has limitations for poor sensitivity and specificity [129]. First, Li et al. characterized circRNAs using RNA-seq analyses of MHCC-LM3 liver cancer cells and cell-derived exosomes [130]. In total, 6751 circRNAs were found in cell-derived exosomes. This number is at least two-fold higher in exosomes compared to that in the donor cells. Moreover, exosomal circRNA_100284 was found in the serum of arsenite-exposed patients [131]. Further functional analysis revealed that exosomal circRNA_100284, derived from malignant-transformed L-02 cells after arsenite exposure enhanced cell cycle and proliferation of normal liver cells and induced malignant transformation of non-transformed cells by acting as a sponge for miRNA-217. In addition, Wang et al. identified exosomal circPTGR1 in a metastatic liver cancer cell line, LM3, which promotes hepatocellular carcinoma metastasis via the miR449a–MET pathway [132]. Notably, some studies demonstrated that adipose-derived exosomal circRNA, circ-DB, promotes HCC cell growth and reduces DNA damage by suppressing miR-34a and activating the USP7/Cyclin A2 signaling pathway [133]. Thus, capitalizing on the high stability and tissue-specificity of exosomal circRNAs, might provide promising cancer biomarkers for early diagnosis and prognosis in HCC patients.

Accumulating evidence has revealed the involvement of ncRNAs in HCC chemo-drug resistance. Sorafenib targeting multiple receptor tyrosine kinases (RTKs) is currently an effective first-line therapy for HCC. However, sorafenib resistance is frequently observed during HCC treatment [134]. The lncRNA TUC338 has been known to be involved in the development of HCC and sorafenib resistance [135]. Enhanced TUC338 expression was observed in both HCC tissues and cell lines. Sorafenib silenced TUC338 in sensitized HepG2 HCC cells, which was accompanied by increased expression of RASAL1. Inversely, intratumoral delivery of siTUC338 could also restore the sorafenib treatment response in HepG2/Sor xenografts in vivo. Moreover, extracellular vesicle-enriched linc-VLDLR is also involved in sorafenib resistance of HCC cells [136]. Depletion of linc-VLDLR led to a reduction of the drug-resistant protein ABCG2 (ATP-binding cassette, subfamily G member 2), which resulted in the suppression of HCC cell proliferation and cell cycle arrest in G1/S. However, elevation of ABCG2 protein inhibited sorafenib-induced cell death by VLDLR knockdown. Recently, aberrant expression of circRNAs has also been identified in sorafenib-resistant HCC cells [137]. Based on high-throughput RNA-sequencing and analysis with CIRI (V2.0) software [138,139], 1,717 and 582 differentially expressed circRNAs were identified in sorafenib-resistant Huh7-S and HepG2-S HCC cells compared to parental HCC cells, respectively. In further gene ontology and pathway analyses, downregulated hsa_circ_0006294 and hsa_circ_0035944 expression was observed in sorafenib-resistant HCC cells.

## 5. Conclusions

In this review, we describe lncRNAs/circRNAs as sponges for regulating miRNAs and their target genes in HCC. We highlight examples of ncRNAs participating in regulatory networks and how they contribute to cancer malignancy. However, the understanding of ceRNA network regulation in the ncRNA field in HCC remains limited, because many HCC-related miRNAs and ncRNAs have not been identified. Therefore, further studies are necessary to elucidate the functions and mechanisms of HCC-related ceRNAs. Furthermore, the lncRNA, circRNA, miRNA, and miRNA target mRNAs involved in the ceRNA network can be potential therapeutic targets and diagnostic markers for HCC. Results from studies on ncRNAs in cancer are very promising. However, classic biomarkers and derived scores continue to be used as a golden standard in the early detection of HCC. Therefore, large prospective studies for the validation of diagnosis using ncRNA biomarkers should be conducted.

## Figures and Tables

**Figure 1 cancers-12-02622-f001:**
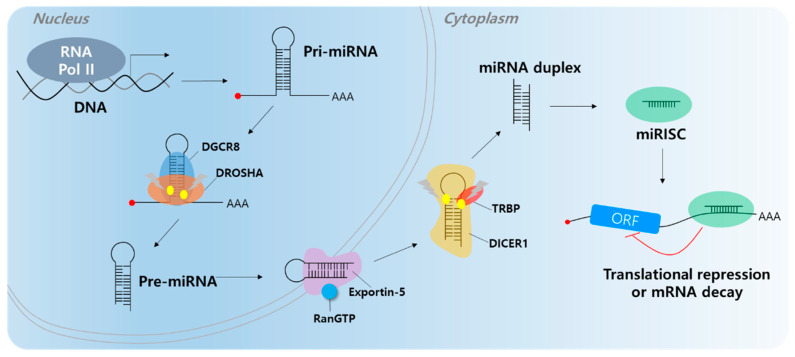
MicroRNA biogenesis. The primary miRNA (pri-miRNA) is transcribed by RNA polymerase II (RNA Pol II). The microprocessor complex, Drosha and DiGeorge Syndrome Critical Region 8 (DGCR8), cleaves the pri-miRNA to produce the precursor miRNA (pre-miRNA). The pre-miRNA translocates to the cytoplasm in an Exportin-5/RnaGTP-dependent manner. TAR RNA-binding protein (TRBP) and Dicer1 cleave the pre-miRNA to produce the mature miRNA duplex. The 5p or 3p of the miRNA duplex is loaded into the miRNA-induced silencing complex (miRISC). Finally, the miRISC binds to target mRNAs to induce translational repression or mRNA decay.

**Figure 2 cancers-12-02622-f002:**
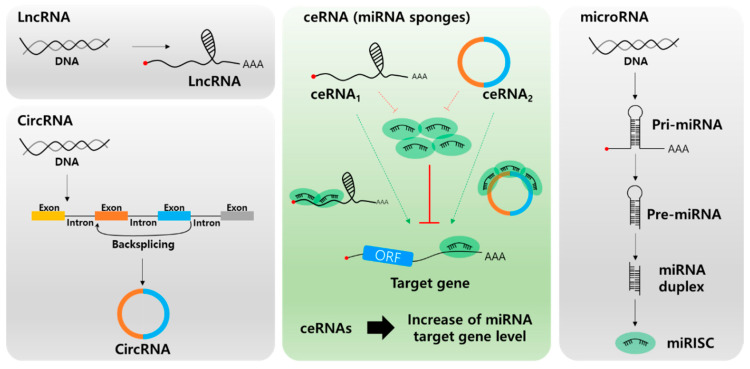
Biogenesis of long non-coding RNA (lncRNA, top left), circular RNA (circRNA, bottom left), microRNA (miRNA, right), and the role of lnc/circRNA as competing endogenous RNAs (ceRNAs, center). Different types of ceRNAs including lncRNA and circRNA regulate the miRNA target mRNA expression by competing for miRNA binding.

**Figure 3 cancers-12-02622-f003:**
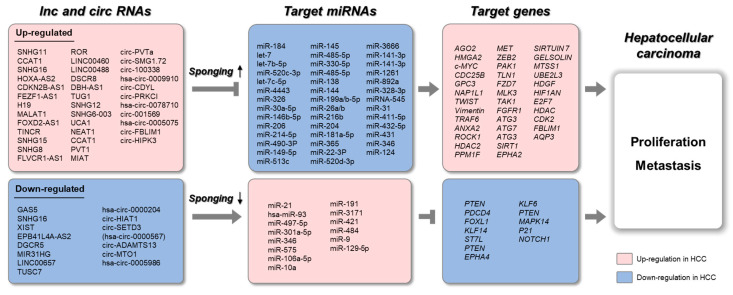
Schematic representation of lnc/circRNAs serving as ceRNAs in HCC. The upregulated long non-coding RNAs (lncRNAs) and circular RNAs (circRNAs) sponge target microRNAs (miRNAs), which results in increased expression of target genes. The downregulated lncRNAs and circRNAs sponge target miRNAs, which results in decreased expression of target genes.

**Table 1 cancers-12-02622-t001:** Long non-coding RNA and microRNA networks in HCC.

lncRNAs	Target miRNA	Target Genes of miRNA	Function	Reference
Overexpression of lncRNAs in HCC				
SNHG11	miR-184	*AGO2*	HCC proliferation, migration, autophagy	[55]
CCAT1	let-7	*HMGA2, c-MYC*	HCC proliferation, migration	[56]
SNHG16	let-7b-5p	*CDC25B*	Cell cycle, migration/invasion	[57]
HOXA-AS2	miR-520c-3p	*GPC3*	Cell cycle, apoptosis, migration/invasion	[26]
CDKN2B-AS1	let-7c-5p	*NAP1L1*	Cell cycle, apoptosis, migration/invasion	[58]
FEZF1-AS1	miR-4443		HCC proliferation, metastasis	[59]
H19	miR-326	*TWIST*	HCC proliferation, metastasis	[60]
MALAT1	miR-30a-5p miR-146b-5p	*Vimentin* *TRAF6*	HCC proliferation, metastasis	[61] [62]
FOXD2-AS1	miR-206	*ANXA2*	HCC proliferation, metastasis	[63]
TINCR	miR-214-5p	*ROCK1*	HCC proliferation, metastasis	[64]
SNHG15	miR-490-3P	*HDAC2*	HCC proliferation, metastasis	[65]
SNHG8	miR-149-5p	*PPM1F*	HCC proliferation, metastasis	[66]
FLVCR1-AS1	miR-513c	*MET*	HCC proliferation, metastasis	[67]
ROR	miR-145	*ZEB2*	HCC metastasis	[68]
LINC00460	miR-485-5p	*PAK1*	HCC proliferation, angiogenesis	[69]
LINC00488	miR-330-5p	*TLN1*	HCC proliferation	[70]
DSCR8	miR-485-5p	*FZD7*	Wnt/β-catenin signaling pathway	[71]
DBH-AS1	miR-138		FAK/Src/ERK pathway	[72]
TUG1	miR-144		JAK2/STAT3 pathway	[73]
SNHG12	miR-199a/b-5p	*MLK3*	NF-κB pathway	[74]
SNHG6-003	miR-26a/b	*TAK1*	p38 pathway	[75]
UCA1	miR-216b	*FGFR1*	FAFR1/ERK signaling pathway	[76]
NEAT1	miR-204	*ATG3*	HCC autophagy process	[77]
CCAT1	miR-181a-5p	*ATG7*	HCC autophagy process	[78]
PVT1	miR-365	*ATG3*	HCC autophagy process	[79]
MIAT	miR-22-3P miR-520d-3p	*SIRT1* *EPHA2*	Cellular senescence HCC proliferation, metastasis	[80] [81]
Downregulation of lncRNAs in HCC				
GAS5	miR-21	*PTEN*	Tumor suppressor	[82]
SNHG16	has-miR-93		Tumor suppressor, 5-FU chemoresistance	[83]
XIST	miR-497-5p	*PDCD4*	Tumor suppressor	[84]
EPB41L4A-AS2	miR-301a-5p	*FOXL1*	Tumor suppressor	[85]
DGCR5	miR-346	*KLF14*	Tumor suppressor	[86]
MIR31HG	miR-575	*ST7L*	Tumor suppressor	[87]
LINC00657	miR-106a-5p	*PTEN*	Tumor suppressor	[88]
TUSC7	miR-10a	*EPHA4*	Tumor suppressor	[89]

**Table 2 cancers-12-02622-t002:** Circular RNA and microRNA networks in HCC.

CircRNAs	Target miRNA	Target Genes of miRNA	Function	Reference
Overexpression of lncRNAs in HCC				
circ-PVTa	miR-3666	*SIRTUIN 7*	HCC proliferation	[90]
circ-SMG1.72	miR-141-3p	*GELSOLIN*	HCC invasion	[94]
circ-100338	miR-141-3p	*MTSS1*	hepatitis B-related HCC progression	[95]
has-circ-0009910	miR-1261	*UBE2L3*	HCC progression	[96]
circ-CDYL	miR-892a, miR-328-3p	*HDGF*, *HIF1AN*	Early stage HCC progression	[97]
circ-PRKCI	miRNA-545	*E2F7*	HCC proliferation	[98]
has-circ-0078710	miR-31	*HDAC*, *CDK2*	HCC progression	[99]
circ-001569	miR-411-5p miR-432-5p	unknown	HCC proliferation, metastasis	[100]
has-circ-0005075	miR-431	unknown	HCC proliferation, metastasis	[101]
circ-FBLIM1	miR-346	*FBLIM1*	HCC progression	[102]
circ-HIPK3	miR-124	*AQP3*	HCC proliferation, metastasis	[103]
Downregulation of lncRNAs in HCC				
has-circ-0000204	miR-191	*KLF6*	HCC proliferation	[104]
circ-HIAT1	miR-3171	*PTEN*	HCC proliferation	[105]
circ-SETD3 (has-circ-0000567)	miR-421	*MAPK14*	HCC proliferation	[106]
circ-ADAMTS13	miR-484	unknown	HCC proliferation	[107]
circ-MTO1	miR-9	*P21*	HCC progression	[108]
has-circ-0005986	miR-129-5p	*NOTCH1*	HCC biomarker	[109]

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
