# Peer review of "Epigenetic Associations between lncRNA/circRNA and miRNA in Hepatocellular Carcinoma"

_cancers, 2020, doi:10.3390/cancers12092622_

Round 1

Reviewer 1 Report

The manuscript entitled "Epigenetic associations between lncRNAs/circRNAs and miRNAs in hepatocellular carcinoma" by Han et al. is a study of lncRNAs/circRNAs as sponges for miRNAs function in hepatocellular carcinoma. It may be important in reviewing recent findings on the role of circRNAs. However, the reviewer feels that this manuscript only describes the functions of individual lncRNAs/circRNAs and does not aid in a comprehensive understanding of lncRNAs/circRNAs in hepatocellular carcinoma. The authors conclude that we do not yet have a good understanding of how the ceRNA network is regulated in the lncRNA field in hepatocellular carcinoma. If the authors believe this is the case, specific examples will help to clarify future challenges.

Author Response

>> Response: The ceRNA network serves as a good model to understand cancer progression. Importantly, increasing evidence of ceRNAs role by experimental validation may help unravel the clues to the HCC progression. However, there remains a lack of information to fully understand the role of ceRNAs in HCC progression, as there are many undiscovered miRNAs as well as ncRNAs in HCC. Thus, we have revised the sentence as follows (line 364):

However, the understanding of ceRNA network regulation in ncRNA field in HCC remains limited, because many HCC-related miRNAs and ncRNAs have not been identified.

Reviewer 2 Report

I read with great interest your paper on epigenetic associations between lncRNA/circRNA and miRNA in hepatocellular carcinoma. It provides a good overview of lncRNA and circRNA functions and their association in HCC processes.

I would like to suggest a correction of the English grammar and review of the text to avoid overusage of certain word for better readibility. Moreover, it is advisable to use the same notations of abbreviations in all parts of the text - either circ-RNA or circRNA.

There was a limited diversity in the choise of references for the paper and we suggest broadening the spectrum by including references arrising from authors from all parts of the world provided the thematical fit.

Author Response

I read with great interest your paper on epigenetic associations between lncRNA/circRNA and miRNA in hepatocellular carcinoma. It provides a good overview of lncRNA and circRNA functions and their association in HCC processes.

I would like to suggest a correction of the English grammar and review of the text to avoid overusage of certain word for better readibility. Moreover, it is advisable to use the same notations of abbreviations in all parts of the text - either circ-RNA or circRNA.

>> Response: Thank you very much for your comment. The manuscript have been carefully edited to improve punctuation, grammar, native tone, and readability by highly qualified native English speaking editor at Editage (www.editage.com). In addition, we checked the abbreviation in all parts of the manuscript.

There was a limited diversity in the choise of references for the paper and we suggest broadening the spectrum by including references arrising from authors from all parts of the world provided the thematical fit.

>> Response: Thank you for your suggestion. We have updated the references in introduction.

[22] Yao, R.W.; Wang, Y.; Chen, L.L. Cellular functions of long noncoding RNAs. Nat Cell Biol 2019, 21, 542-551, doi:10.1038/s41556-019-0311-8.

[24] Schmitt, A.M.; Chang, H.Y. Long Noncoding RNAs in Cancer Pathways. Cancer Cell 2016, 29, 452-463, doi:10.1016/j.ccell.2016.03.010.

[25] Li, X.; Yang, L.; Chen, L.L. The Biogenesis, Functions, and Challenges of Circular RNAs. Mol Cell 2018, 71, 428-442, doi:10.1016/j.molcel.2018.06.034.

[27] Zhang, X.; Wang, S.; Wang, H.; Cao, J.; Huang, X.; Chen, Z.; Xu, P.; Sun, G.; Xu, J.; Lv, J., et al. Circular RNA circNRIP1 acts as a microRNA-149-5p sponge to promote gastric cancer progression via the AKT1/mTOR pathway. Mol Cancer 2019, 18, 20, doi:10.1186/s12943-018-0935-5.

[41] Peng, Y.; Croce, C.M. The role of MicroRNAs in human cancer. Signal Transduct Target Ther 2016, 1, 15004, doi:10.1038/sigtrans.2015.4.

[42] Jansson, M.D.; Lund, A.H. MicroRNA and cancer. Mol Oncol 2012, 6, 590-610, doi:10.1016/j.molonc.2012.09.006.

Reviewer 3 Report

The manuscript submitted by Han et al. enitled: „Epigenetic associations between lncRNA/circRNA and miRNA in hepatocellular carcinoma“   gives an overview on the current knowledge about the role of non-coding RNAs in HCC detection and prediction. In addition the paper discusses the potential role of ncRNAs in chemo-drug- resistance which is a clinically relevant aspect. The paper gives a detailed overview on the topic and is written comprehensively.

Major issues:

In the chapter 4  the authors report that UCA 1 could distinhguish between healthy controls (sure, that it did not use non-HCC patients with liver disease?) and HCC with  an  AUC for level of 0.902 in HCC detection , however it remains unclear, whether those are BCLC 0/A or any BCLC stage patients. If any stage, the AUC performs less good than other biomarker based  validated models, such as GALAD-score. Here the ncRNA data  should  be discussed in the light of recent  publications reaching AUCs for GALAD up to 0.97 in the overall HCC population and up to 0.92 in the BCLC early stages.

Minor issues

The page numbers are not in correct order

The figure legends need to be more detailed, many abbreviations in the figures are not explained in detail.

In the introduction the authors state, that NAFLD represents an HCC risk-factor, which has to be refined a bit. The majority of HCC develop under the inflammatory stage of NAFLD: NASH. Therefore I would suggest to substitute NAFLD with NASH.

The  conclusion needs to state in an introductory sentence, that despite very promising results the ncRNA data did not undergo prospective validation and therefore classical biomarkers and deriven scores are still state oft he art in HCC early detection.

Author Response

The manuscript submitted by Han et al. entitled: „Epigenetic associations between lncRNA/circRNA and miRNA in hepatocellular carcinoma“ gives an overview on the current knowledge about the role of non-coding RNAs in HCC detection and prediction. In addition the paper discusses the potential role of ncRNAs in chemo-drug- resistance which is a clinically relevant aspect. The paper gives a detailed overview on the topic and is written comprehensively.

Major issues:

In the chapter 4 the authors report that UCA 1 could distinguish between healthy controls (sure, that it did not use non-HCC patients with liver disease?) and HCC with an AUC for level of 0.902 in HCC detection , however it remains unclear, whether those are BCLC 0/A or any BCLC stage patients. If any stage, the AUC performs less good than other biomarker based validated models, such as GALAD-score. Here the ncRNA data should be discussed in the light of recent publications reaching AUCs for GALAD up to 0.97 in the overall HCC population and up to 0.92 in the BCLC early stages.

>> Response: We would like to thank the reviewer for this particularly important comment. We have elaborated on the importance of multiple score models in biomarker evaluation in our revised manuscript and have cited 2 references (#115 and #116) in our revised submission as follows (line 306):

However, in order to apply this criterion clinically, additional evaluations are required. GALAD (which includes gender, age, AFP-L3, alpha-fetoprotein, and des-carboxy-prothrombin) and BALAD (which includes bilirubin, albumin, AFP-L3, alpha-fetoprotein, and des-carboxy-prothrombin) score calculations may be particularly useful [115,116].

References

[115] Roberts, L.R. Current status of the galad and balad biomarker models for hepatocellular carcinoma. Gastroenterol Hepatol (N Y) 2019, 15, 672-675.

[116] Best, J.; Bechmann, L.P.; Sowa, J.P.; Sydor, S.; Dechene, A.; Pflanz, K.; Bedreli, S.; Schotten, C.; Geier, A.; Berg, T., et al. Galad score detects early hepatocellular carcinoma in an international cohort of patients with nonalcoholic steatohepatitis. Clin Gastroenterol Hepatol 2020, 18, 728-735 e724.

Minor issues

The page numbers are not in correct order

>> Response: We have ensured that the page numbers are in correct order when using the Word format.

The figure legends need to be more detailed, many abbreviations in the figures are not explained in detail.

>> Response: We have revised the figure legends as follows:

Figure 1. MicroRNA biogenesis. The primary miRNA (pri-miRNA) is transcribed by the RNA polymerase II (RNA Pol II). The microprocessor complex, Drosha and DiGeorge Syndrome Critical Region 8 (DGCR8), cleaves the pri-miRNA to produce the precursor miRNA (pre-miRNA). The pre-miRNA is translocated to the cytoplasm in an Exportin-5/RnaGTP-dependent manner. The TAR RNA-binding protein (TRBP) and Dicer1 cleave the pre-miRNA to produce the mature miRNA duplex. The 5p or 3p of miRNA duplex is loaded into the miRNA-induced silencing complex (miRISC). Finally, miRISC binds to target mRNAs to induce translational repression or mRNA decay.

Figure 2. Biogenesis of long non-coding RNA (lncRNA, top left), circular RNA (circRNA, bottom left), microRNA (miRNA, right), and the role of lnc/cirRNA as competing endogenous RNA (ceRNA, center). Different types of ceRNAs including lncRNA and circRNA regulate the miRNA target mRNA expression by competing for miRNA binding.

Figure 3. Schematic representation of lnc/circRNAs serving as ceRNA in HCC. The up-regulated long non-coding RNAs (lncRNAs) and circular RNAs (circRNAs) sponge target microRNAs (miRNAs), which result in increased expression of target genes. The down-regulated lncRNAs and circRNAs sponge target miRNAs, which result in decreased expression of target genes.

In the introduction the authors state, that NAFLD represents an HCC risk-factor, which has to be refined a bit. The majority of HCC develop under the inflammatory stage of NAFLD: NASH. Therefore I would suggest to substitute NAFLD with NASH.

>> Response: According to reviewer’s helpful opinion, we have mentioned “non-alcoholic steatohepatitis (NASH)” in the introduction as follows (line 35):

Risk factors for HCC are hepatitis B virus (HBV) or hepatitis C virus (HCV) infection, alcohol consumption, nonalcoholic fatty liver disease (NAFLD), non-alcoholic steatohepatitis (NASH), and hereditary diseases [2-7].

The conclusion needs to state in an introductory sentence, that despite very promising results the ncRNA data did not undergo prospective validation and therefore classical biomarkers and deriven scores are still state of the art in HCC early detection.

>> Response: We appreciate the reviewer’s valuable suggestion. As suggested, we have revised the following sentences in the conclusion section (line 369):

The results from the studies on ncRNA roles in cancer are very promising; however, classic biomarkers and derived scores continue to be used as a golden standard in the early detection of HCC. Therefore, large prospective studies for the validation of diagnosis using ncRNA biomarkers should be conducted.

Round 2

Reviewer 3 Report

The authors responded to all the Reviewers Points appropriately. Therefore the Reviewer esteems the manuscript can be published in its current form. 

Author Response

Thank you very much for your comment.